# Risk Factors for Postoperative Deep Infection after Instrumented Spinal Fusion Surgeries for Degenerative Spinal Disease: A Nationwide Cohort Study of 194,036 Patients

**DOI:** 10.3390/jcm11030778

**Published:** 2022-01-31

**Authors:** Jihye Kim, Tae-Hwan Kim

**Affiliations:** 1Division of Infection, Department of Pediatrics, Kangdong Sacred Heart Hospital, Hallym University College of Medicine, Seoul 05355, Korea; jihyewiz17@kdh.or.kr; 2Spine Center, Department of Orthopedics, Hallym University Sacred Heart Hospital, Hallym University College of Medicine, Anyang 14068, Korea

**Keywords:** postoperative infection, deep infection, spine surgery, spinal fusion, instrumentation, risk factor

## Abstract

Previous studies to identify risk factors for postoperative deep infection following instrumented spinal fusion surgery for degenerative spinal disease are based on insufficient information and have limited use in clinical practice. This study aims to fill this gap by assessing the risk factors and their adjusted relative risks through a comprehensive analysis, including all core information. In this nationwide, population-based, cohort study, data were obtained from the Korean National Health Insurance claims database between 1 January 2014, and 31 December 2018. This study included a cohort of 194,036 patients older than 19 years, who underwent instrumented spinal fusion surgery for degenerative spinal disease. We divided this population into cases (patients with postoperative deep infection) and controls (patients without postoperative deep infection); risk factors for postoperative deep infection were determined by multivariable analysis. The definition of postoperative deep infection varied, and sensitivity analyses were performed according to each definition. The estimates of all the statistical models were internally validated using bootstrap samples. The study included 767 patients (0.39%) with postoperative deep spinal infections and 193,269 controls. The final multivariable model identified the following variables as significant risk factors for postoperative deep infection: age between 60–69 years (OR = 1.6 [1.1–2.3]); age between 70–79 years (OR = 1.7 [1.2–2.5]); age > 80 years (OR = 2.1 [1.3–3.2]); male sex (OR = 1.7 [1.5–2.0]); rural residence (OR = 1.3 [1.1–1.5]); anterior cervical approach (OR = 0.2 [0.1–0.3]); posterior cervical approach (OR = 0.5 [0.2–1.0]); multiple approaches (OR = 1.4 [1.2–1.6]); cerebrovascular disease (OR = 1.5 [1.2–1.8]); peripheral vascular disease (OR = 1.3 [1.1–1.5]); chronic pulmonary disease (OR = 1.2 [1.0–1.4]); rheumatologic disease (OR = 1.6 [1.3–2.1]); liver disease (OR = 1.4 [1.1–1.7]); diabetes (OR = 1.5 [1.3–1.7]); hemiplegia or paraplegia (OR = 2.2 [1.5–3.3]); allogenous transfusion (OR = 1.6 [1.3–1.8]); and use of systemic steroids over 2 weeks (OR = 1.5 [1.1–2.0]). Our results, which are based on homogenous patient groups, provide clinicians with an acceptable tool for comprehensive risk assessment of postoperative deep infection in patients who will undergo instrumented spinal fusion surgery for degenerative spinal disease.

## 1. Introduction

Degenerative spinal disease is one of the most common chronic disorders affecting members of an aging population [1], lowering their quality of life with chronic pain and physical disability. Most patients with degenerative spinal disease can be treated conservatively with pain medication. However, the severity of the disease is closely associated with age; in an aging population, a significant number of people undergo surgery for advanced disease, and various types of spinal instruments are inevitably used to support their weak spinal structures before or after surgery. Postoperative infection in these patients is one of the most devastating postoperative complications.

Studies do not consider age a consistent risk factor for infection after spinal surgery. However, patients’ comorbidities and the level of surgical invasiveness are believed to be associated with a higher risk of postoperative spinal infection [2,3]. Because members of an aging population are more predisposed to developing medical comorbidities than those of a younger population, the incidence of postoperative spine infection is also expected to be increased in them.

Postoperative deep infection occurring in patients who have undergone instrumented spinal fusion surgeries can cause mortality or disastrous complications resulting from long-term antibiotic use and immobilization. These infections can also necessitate further surgeries including removal of a previously inserted instrument to control the infection or additional instrumentation for stabilization [4]. Therefore, when spine surgeons plan instrumented spinal fusion surgery for patients with degenerative spinal disease, identifying those at high risk of postoperative deep infection is a prerequisite to avoiding postoperative complications.

Studies have attempted to identify the risk factors for postoperative spinal infections. However, there were several limitations. Most of the studies were based on a limited number of patients, which was not representative of the general population [3,5]. There have also been several studies with large sample sizes [6,7,8,9,10,11,12,13,14,15]; however, these studies had methodological limitations. First, the target diseases for which spinal surgery was performed were not clearly outlined or analyzed individually. Second, the patient population was heterogeneous, including patients who underwent all types of spinal surgery, regardless of spinal instrumentation. The risk factors for postoperative infection in patients with different diagnoses and the surgical methods varied. Third, postoperative spinal infection was not properly defined or classified. The clinical outcomes of superficial and deep wound infections, including those of the vertebral body or intervertebral disc, are different and should be analyzed separately. Fourth, several types of core clinical information that influence the occurrence of postoperative infection, such as comorbidities and surgical approach, were not comprehensively evaluated. Finally, events that occurred between the index spinal surgery and the diagnosis of postoperative spinal infection, such as epidural injection, were not considered.

The purpose of our study is to investigate the risk factors for postoperative deep infection among patients who underwent instrumented spinal fusion surgery for degenerative spinal disease and to present their relative risks through a comprehensive analysis, including all possible core information. To overcome these limitations, our study used a domestic national claims database that covers the entire population.

## 2. Patients and Methods

### 2.1. Database

In this nationwide, population-based, cohort study, data were obtained from the Health Insurance Review and Assessment Service (HIRA) (which contains data from the Korean National Health Insurance (NHI)) and the National Medical Aid (NMA) databases. The NHI covers approximately 97% of the population, and the NMA program covers the remaining 3%. The HIRA database reviews all data from the NHI and NMA. Therefore, the HIRA database contains all inpatient and outpatient data from hospitals and community clinics in Korea, making a nationwide cohort study feasible. The diagnostic codes were assigned according to a modified version of the tenth revision of the International Classification of Disease (ICD-10) and the seventh revision of the Korean Classification of Disease (KCD-7). Drug use under diagnosis was identified using the Anatomical Therapeutic Chemical and HIRA general name codes. This study was approved by the institutional review board of our hospital (IRB No. 2020-03-009-001).

### 2.2. Study Patients

We included patients who underwent instrumented spinal fusion surgeries due to degenerative spinal disease, between 1 January 2012, and 31 December 2018 and those aged >19 years. Degenerative spinal diseases were identified using the following codes: M48.0 (spinal stenosis), M43.1 (spondylolisthesis), M43.0 (spondylolysis), M47.1 (spondylosis with myelopathy), M47.2 (spondylolysis with radiculopathy), and M50 (cervical disc disorder). Surgical approaches were identified and classified using the following electronic data interchange codes: anterior cervical approach (N2461, N0464, N2463), posterior cervical approach (N2467, N2468, N0467, N2469), anterior thoracic approach (N0465, N2464, N2465, N2466), posterior thoracic approach (N0468), anterior lumbar approach (N0466, N1466), posterior lumbar approach (N0469, N1460, N1469, N2470), and multiple approach, including two or more of the aforementioned approaches. We excluded patients who were treated under the ICD-10 codes of spine infection or had undergone previous spinal surgeries between 2012 and 2013 (wash-out period, Figure 1). We also excluded patients who underwent another clean spine surgery (without ICD-10 codes of spine infection) within one year of the index surgery.

### 2.3. Definitions

We defined postoperative deep infection as any infection occurring within one year of the index spinal fusion surgery. A minimum follow-up period of one year was mandatory for inclusion in the study. To determine a postoperative deep spinal infection, an infectious event was expected to have required intravenous antibiotic therapy over 4 weeks for the defined ICD-10 codes (Appendix A) [16]. Postoperative deep spinal infection was identified using the following ICD-10 codes [8,17]: intraspinal abscess (G06.1); osteomyelitis of vertebra (M46.2); discitis (M46.3, M46.4); other or unspecified infection (M46.5, M46.8, M46.9, M49.2, M49.3); and unspecified extradural and subdural abscess (G06.2). In addition, to exclude infections associated with additional postoperative procedures after the index surgery, we excluded patients who underwent additional local invasive spinal procedures (Figure 1) within 90 days prior to the diagnosis of spinal infection (Figure 2) [17].

### 2.4. Covariates

Preexisting comorbidities relevant to postoperative deep infection were identified according to the ICD-10 codes (Appendix A), within one year prior to the index surgery, based on the Charlson comorbidity index (CCI). The CCI score is the sum of the weighted scores of each comorbidity and has shown good agreement with ICD-10 codes [18,19]. Data regarding transfusion (allogenous or autologous) for the index surgery (Appendix A) and use of steroids (Appendix A) and immunosuppressants within two weeks before and after the index surgery were retrieved [20]. The type of hospital and region of residence were defined in accordance with previous studies [20,21].

### 2.5. Statistical Analysis

Data are reported as mean ± standard deviation for numerical variables and as numbers and frequencies (%) for categorical variables. Logistic regression models were used to construct prediction models for postoperative deep infections. All significant independent variables (*p* < 0.05) from the univariate analysis were included in the first multivariable model (Model 1) and subsequently chosen by backward stepwise selection (Model 2, final model). Multicollinearity between covariates was tested using the variance inflation factor. The performance of the prediction model was assessed by the area under the receiver operating characteristic curve, for discriminative ability, and by the Hosmer–Lemeshow goodness-of-fit statistics, for calibration.

Postoperative deep infection was additionally defined using the following methods (Table 1): shorter (2 weeks or more) and longer (6 weeks or more) duration of therapeutic intravenous antibiotics for postoperative deep spine infection, and limiting postoperative deep infection as an infection that occurred within 60 days of the index surgery. Sensitivity analysis was performed according to the three definitions. The estimates of all statistical models were internally validated with a relative bias based on 1000 bootstrapped samples. Data extraction and statistical analysis were performed using SAS Enterprise Guide 6.1 (SAS Institute, Cary, NC, USA).

## 3. Results

Among 216,033 patients who underwent instrumented spinal fusion surgery for degenerative spinal disease between 2014 and 2018, 3060 were initially identified as having the ICD-10 codes for postoperative deep spinal infection (Figure 1). Among them, 262 (8.6% of 3060) who had undergone additional local invasive spinal procedures within 90 days prior to the diagnosis of deep infection, were excluded (Figure 1). Of the remaining 2798 patients, 2031 (72.6%) received intravenous antibiotics for less than 4 weeks and were excluded. Finally, our study included 194,036 patients divided into two cohorts: 767 and 193,269 patients with and without postoperative deep spinal infection, respectively. The median interval between the index surgery and the occurrence of infection was 54 days (interquartile range, 29–123 days).

### 3.1. Incidence of Postoperative Deep Infection

Table 1 presents the incidence of postoperative spinal infections according to the four definitions. The incidence rate of postoperative infection decreased as the defined duration of therapeutic intravenous antibiotics increased: it was 0.5%, 0.39%, 0.3% with antibiotics over 2, 4, and 6 weeks, respectively (Table 1). Regardless of the definition, the incidence of postoperative spinal infection decreased during the study period (Figure 3).

### 3.2. Comparison of the Two Patient Cohorts with and without Postoperative Deep Infection: Univariable Analysis

Baseline characteristics of the patients in the two cohorts are presented in Table 2. Male sex, older age groups, rural residence, higher hospital volume, CCI score, and various comorbidities were associated with an increased risk of postoperative deep infection. The treatment profiles of the two cohorts are presented in Table 3. Surgical approach, cage use, allogeneic transfusion, and prolonged use of systemic steroids (over 2 weeks) were associated with an increased risk of postoperative deep infection.

### 3.3. Risk Factors for Postoperative Deep Infection after Spinal Fusion: Multivariable Analysis with Bootstrap Validation

Model 1 (Table 4) was created from the multivariable analysis, while Model 2 was established using the variables chosen from a backward stepwise selection (Table 5). Model 2 included the following variables: age between 60–69 years (OR = 1.6); age between 70–79 years (OR = 1.7); age >80 years (OR = 2.1); male sex (OR = 1.7); rural residence (OR = 1.3); anterior cervical approach (OR = 0.2); posterior cervical approach (OR = 0.5); multiple approaches (OR = 1.4); cerebrovascular disease (OR = 1.5); peripheral vascular disease (OR = 1.3); chronic pulmonary disease (OR = 1.2); rheumatologic disease (OR = 1.6); liver disease (OR = 1.4); diabetes (OR = 1.5); hemiplegia or paraplegia (OR = 2.2); allogeneic transfusion (OR = 1.6); and use of systemic steroids over 2 weeks (OR = 1.5). Multicollinearity among covariates was low, and all variance inflation factors were less than 2.5. The Hosmer–Lemeshow goodness-of-fit test indicated good calibration (*p* = 0.695), and the area under the receiver operating curve was 0.827.

After bootstrap validation, the relative bias of the estimates in Model 1 was high, between −557.1% and 202.7% (Table 4). After backward selection, the relative bias of the estimate in Model 2 was low, between −7.6% and 10.1% (Table 5). Bootstrap-adjusted odds ratios and confidence intervals for Model 2 are shown in Figure 4.

### 3.4. Sensitivity Analysis

Sensitivity analysis was performed to assess our prediction model according to the different durations of intravenous antibiotics for postoperative deep infection. When defined as an infection requiring over 2-week intravenous antibiotics, 977 patients were identified as having postoperative deep infection. Except for the use of systemic steroids over 2 weeks, all variables in the final model (model 2 in Table 4) remained consistently significant in this prediction model, and the posterior thoracic approach was chosen as a significant predictor (Appendix A). When defined as an infection requiring over 6-week intravenous antibiotics, 584 patients were identified as having postoperative deep infection. Age >80 years and posterior thoracic approach, which were significant predictors for the final model, were not significant predictors for this prediction model (Appendix A). Additionally, end-stage renal disease was identified as a significant predictor.

Sensitivity analysis was also performed when postoperative deep infection was defined as an infection that occurred within 60 days of the index surgery; 417 patients were identified, and the median interval between the index surgery and postoperative infection was 30 days (interquartile range, 21–42 days). Rural residence, cervical posterior approach, chronic pulmonary disease, cerebrovascular disease, hemiplegia or paraplegia, and use of systemic steroids over 2 weeks, which were significant predictors for the final model (Model 2 in Table 4) were not significant predictors for this definition (Appendix A).

For these three additional models, the bootstrap-adjusted odds ratios and confidence intervals are displayed in Figure 5.

## 4. Discussion

The recent incidence of postoperative spinal infection ranges from 0.2% to 16.7% [3,5], and that of deep infection is relatively low (<1.8%) [5,8,22,23]. In our study, 194,036 patients who underwent instrumented spinal fusion surgeries for degenerative spinal disease between 2014 and 2018 were analyzed, and the mean incidence rate of postoperative deep spinal infection in them was 0.39% (767 of 196,329, Table 1). Multivariable analysis identified the following variables as significant risk factors for postoperative deep spinal infection after instrumented spinal fusion surgery for degenerative spinal disease: older age, male sex, rural residence, multiple surgical approach, comorbidities including cerebrovascular disease, peripheral vascular disease, chronic pulmonary disease, rheumatologic disease, liver disease, diabetes, hemiplegia or paraplegia, allogeneic transfusion, and systemic steroid use over 2 weeks (Figure 4). In contrast, the anterior and posterior cervical approaches were associated with a lower risk of infection.

Previous studies have shown conflicting results in evaluating the factors related to the risk of infection such as age [3], and some have shown methodological limitations [6,7,8,9,10,11,12,13,14,15]. Our study has several advantages compared with previous studies with similar research purposes. First, our study is the largest study of this type, and is based on a homogenous group of patients with clear target diseases who underwent spinal fusion surgeries using instrumentation due to degenerative spinal disease. Second, the clinical endpoint (postoperative deep infection) was clearly defined, based on the patients with a minimum of one-year mandatory follow-up after the index surgery (Figure 2). Third, rigorous statistical calculation was performed to overcome the essential limitations of claims databases. Patients were stratified into three categories according to the duration of the postoperative intravenous antibiotics (over 2, 4, and 6 weeks) and into two categories according to the interval between the index surgery and the postoperative infection (within 60 days and all intervals). Sensitivity analysis was performed according to each defined category (Table 5, Appendix A). Bootstrap adjustment was performed for all prediction models (Table 5, Appendix A). Fourth, core clinical information, including precise surgical approach and precise comorbid conditions, could be included using the domestic national claims database covering the entire population. We could also assess all clinical events that occurred between the index surgery and postoperative infection, and exclude patients who underwent additional invasive spinal procedures (Figure 1).

This study has several limitations. First, the claims database is not designed for clinical research; thus, important clinical information, such as laboratory and radiologic data and precise surgical profiles including different sterilization procedures, surgical protocols, and techniques among hospitals or surgeons, was not included in the study, which may have led to bias. However, the consistency of our results was confirmed by a sensitivity analysis using various definitions of postoperative deep infection [24]. Possible discrepancies between the diagnostic codes in the database and the real disease in patients could be potential sources of bias. However, high-revenue invasive procedures and drug use are well documented in the administrative databases [25,26,27] which our study utilized. Several important degenerative diseases, including adult spinal deformity, were not included in our study owing to the limited data capacity for analysis. In our study, patients who underwent multiple surgical approaches showed an increased rate of postoperative deep infection, and those with deformities were also expected to have an increased rate of infection. Future studies including such patients are required.

In conclusion, postoperative deep spinal infection after instrumented spinal fusion surgery for degenerative spinal disease is one of the most devastating complications of spinal surgery. A precise and comprehensive risk assessment for postoperative deep spinal infection is a prerequisite for the clinical success of instrumented spinal fusion surgery for degenerative spinal disease. Previous studies have consistently reported that comorbidities and surgical invasiveness are associated with postoperative spine infection. However, evidence for precise variables such as individual comorbidities or surgical methods was limited [8,28,29,30]. In this regard, our results, which are based on a large number of homogenous patient groups, provide clinicians with an acceptable tool for comprehensive risk assessment of postoperative deep infection in patients who are set to undergo instrumented spinal fusion surgery for degenerative spinal disease.

## Figures and Tables

**Figure 1 jcm-11-00778-f001:**
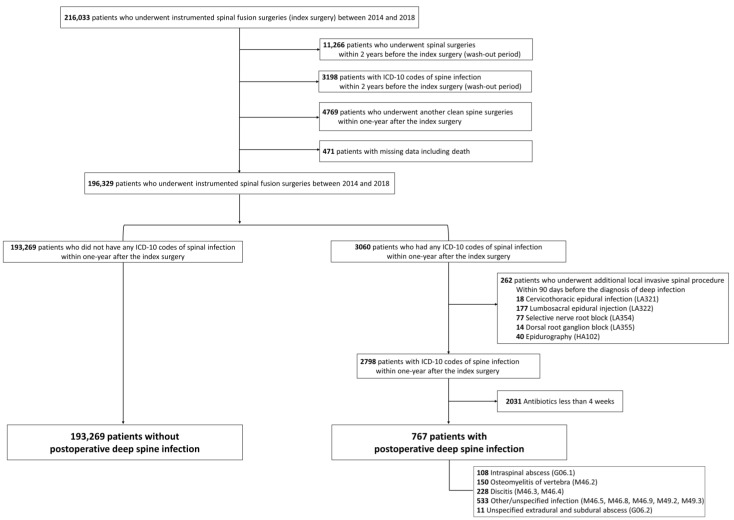
Enrollment of patients in this study. ICD-10: International Classification of Disease.

**Figure 2 jcm-11-00778-f002:**
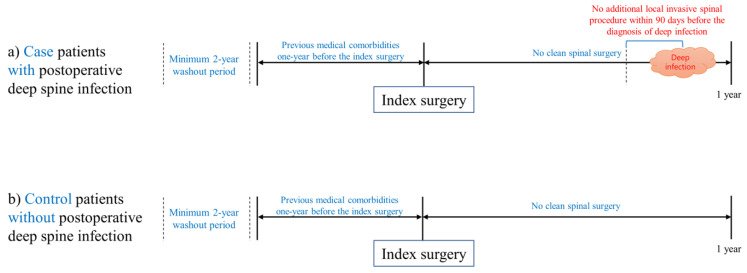
Case and control definitions. (**a**) Cases: patients with postoperative deep spinal infection after instrumented spinal fusion surgery for degenerative spinal disease. (**b**) Controls: patients without postoperative deep spinal infection after instrumented spinal fusion surgery for degenerative spinal disease.

**Figure 3 jcm-11-00778-f003:**
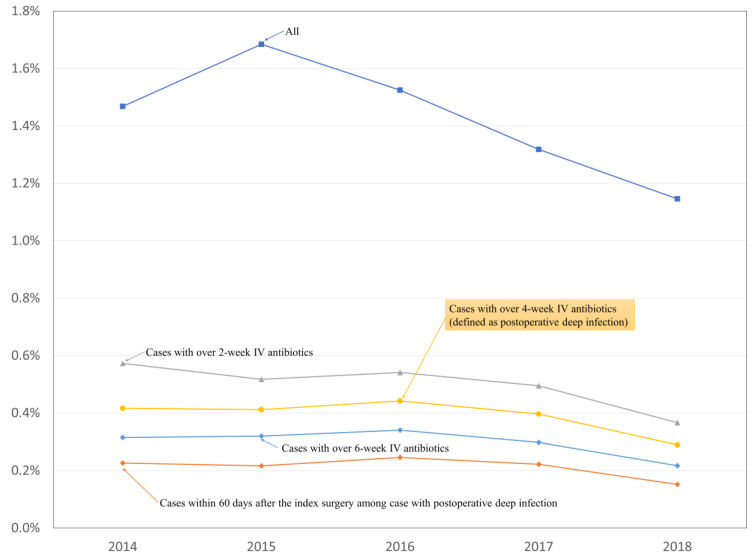
Incidence of postoperative deep spinal infection after instrumented spinal fusion surgery for degenerative spinal disease according to the four definitions. IV: intravenous.

**Figure 4 jcm-11-00778-f004:**
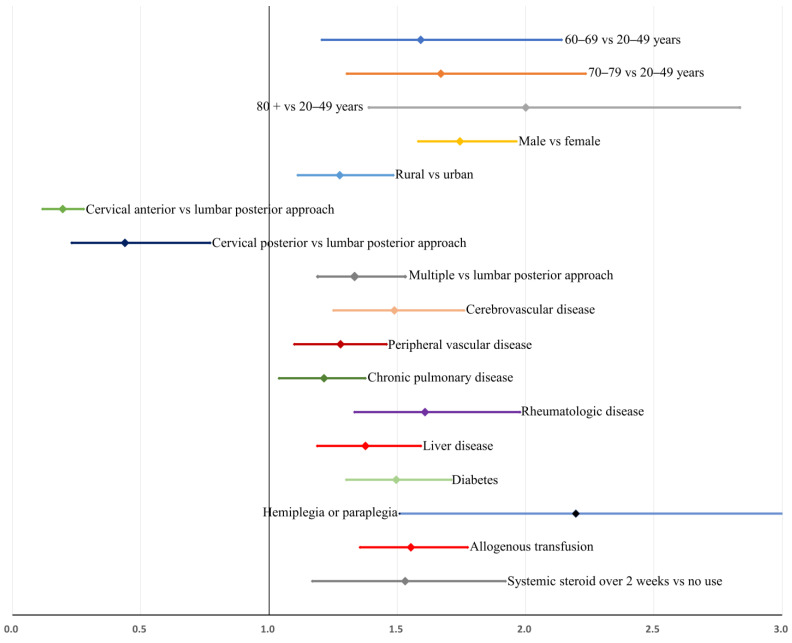
Bootstrap-adjusted odds ratios and their 95% confidence intervals for postoperative deep spinal infection after instrumented spinal fusion surgery for degenerative spinal disease: results from backward, stepwise logistic regression analysis (Model 2 in Table 5).

**Figure 5 jcm-11-00778-f005:**
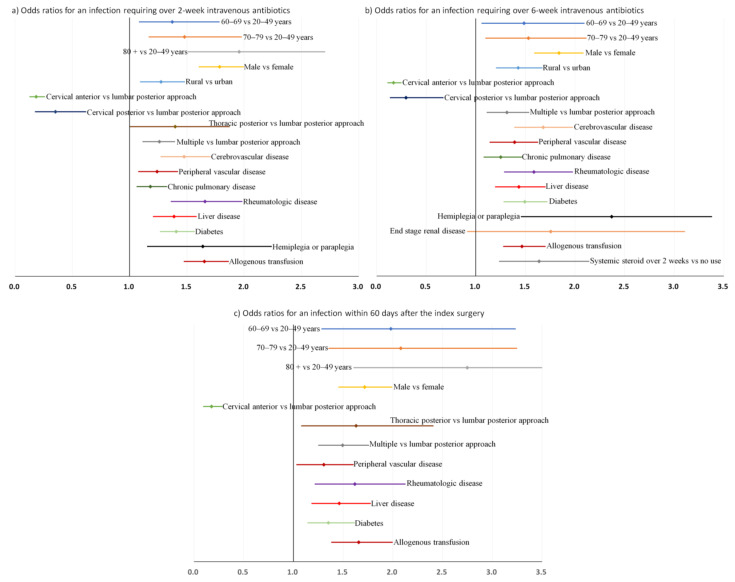
Bootstrap-adjusted odds ratios and their 95% confidence intervals for postoperative deep spinal infection after instrumented spinal fusion surgery for degenerative spinal disease: (**a**) Odds ratios for an infection requiring intravenous antibiotics for 2 weeks (Model 2 in Appendix A), (**b**) Odds ratios for an infection requiring intravenous antibiotics for 6 weeks (Model 2 in Appendix A), and (**c**) Odds ratios for an infection occurring within 60 days of the index surgery (Model 2 in Appendix A).

**Table 1 jcm-11-00778-t001:** Incidence of postoperative deep spine infection after instrumented spinal fusion procedures, according to the four definitions.

Year	Instrumented Spinal Fusion Procedure (*n*)	ALL			Over 2-Week Intravenous Antibiotics			Over 4-Week Intravenous Antibiotics						Over 6-Week Intravenous Antibiotics		
								Occurrence within 60 Days of the Index Surgery			Occurrence within One Year of the Index Surgery (Defined as Postoperative Deep Infection)					
		(*n*)	Incidence (%)	95% CI	(*n*)	Incidence (%)	95% CI	(*n*)	Incidence (%)	95% CI	(*n*)	Incidence (%)	95% CI	(*n*)	Incidence (%)	95% CI
2014	38,426	564	1.47	(1.35–1.59)	220	0.57	(0.50–0.65)	87	0.23	(0.18–0.27)	160	0.42	(0.35–0.48)	121	0.31	(0.26–0.37)
2015	37,877	638	1.68	(1.55–1.81)	196	0.52	(0.45–0.59)	82	0.22	(0.17–0.26)	156	0.41	(0.35–0.48)	121	0.32	(0.26–0.38)
2016	40,277	614	1.52	(1.40–1.64)	218	0.54	(0.47–0.61)	99	0.25	(0.20–0.29)	178	0.44	(0.38–0.51)	137	0.34	(0.28–0.40)
2017	39,608	522	1.32	(1.21–1.43)	196	0.49	(0.43–0.56)	88	0.22	(0.18–0.27)	157	0.40	(0.33–0.46)	118	0.30	(0.24–0.35)
2018	40,141	460	1.15	(1.04–1.25)	147	0.37	(0.31–0.43)	61	0.15	(0.11–0.19)	116	0.29	(0.24–0.34)	87	0.22	(0.17–0.26)
All	196,329	2798	1.43	(1.37–1.48)	977	0.50	(0.47–0.53)	417	0.21	(0.19–0.23)	767	0.39	(0.36–0.42)	584	0.30	(0.27–0.32)

**Table 2 jcm-11-00778-t002:** Comparison of baseline patient characteristics.

Variables	Category	Patients without Postoperative Deep Spine Infection	Patients with Postoperative Deep Spine Infection	Unadjusted Odds Ratio (95% Confidence Interval)	*p*-Value
Number of patients		193,269	767		
Age	20–49	27,635 (14.3)	38 (5.0)	reference	-
	50–59	46,843 (24.2)	120 (15.6)	1.9 (1.3–2.7)	<0.001
	60–69	60,533 (31.3)	272 (35.5)	3.3 (2.3–4.6)	<0.001
	70–79	51,021 (26.4)	285 (37.2)	4.1 (2.9–5.7)	<0.001
	80+	7237 (3.7)	52 (6.8)	5.2 (3.4–7.9)	<0.001
Sex	Male	85,934 (44.5)	390 (50.8)	1.3 (1.1–1.5)	<0.001
	Female	107,335 (55.5)	377 (49.2)	reference	-
Region of residence	Urban	162,894 (84.3)	614 (80.1)	reference	-
	Rural	30,375 (15.7)	153 (19.9)	1.3 (1.1–1.6)	0.001
Type of hospital	Tertiary	53,639 (27.8)	227 (29.6)	1.3 (1.1–1.6)	0.002
	General hospital	49,111 (25.4)	248 (32.3)	1.6 (1.3–1.9)	<0.001
	Others	90,519 (46.8)	292 (38.1)	reference	-
Charlson comorbidity index score	Mean ± SD	1.4 ± 1.5	2.2 ± 1.9	1.3 (1.2–1.3)	<0.001
	0–2	155,084 (80.2)	509 (66.4)	reference	-
	3–5	34,426 (17.8)	207 (27.0)	1.8 (1.6–2.2)	<0.001
	≥6	3759 (1.9)	51 (6.6)	4.1 (3.1–5.5)	<0.001
Comorbidities	Myocardial infarction	1235 (0.6)	7 (0.9)	1.4 (0.7–3.0)	0.342
	Congestive heart failure	7015 (3.6)	46 (6.0)	1.7 (1.3–2.3)	<0.001
	Peripheral vascular disease	23,066 (11.9)	140 (18.3)	1.6 (1.4–2.0)	<0.001
	Cerebrovascular disease	19,483 (10.1)	143 (18.6)	2.0 (1.7–2.5)	<0.001
	Dementia	2650 (1.4)	16 (2.1)	1.5 (0.9–2.5)	0.09
	Chronic pulmonary disease	50,116 (25.9)	262 (34.2)	1.5 (1.3–1.7)	<0.001
	Rheumatologic disease	9902 (5.1)	70 (9.1)	1.9 (1.5–2.4)	<0.001
	Peptic ulcer disease	38,211 (19.8)	181 (23.6)	1.3 (1.1–1.5)	0.008
	Liver disease	23,849 (12.3)	138 (18.0)	1.6 (1.3–1.9)	<0.001
	Mild	23,593 (12.2)	136 (17.7)	1.6 (1.3–1.9)	<0.001
	Moderate to severe	256 (0.1)	2 (0.3)	2.0 (0.5–8.0)	0.339
	Diabetes	57,538 (29.8)	352 (45.9)	2.0 (1.7–2.3)	<0.001
	Uncomplicated diabetes	44,024 (22.8)	272 (35.5)	1.9 (1.6–2.2)	<0.001
	Complicated diabetes	13,514 (7.0)	80 (10.4)	1.5 (1.2–2.0)	<0.001
	Hemiplegia or paraplegia	2395 (1.2)	26 (3.4)	2.8 (1.9–4.1)	<0.001
	Renal disease	4077 (2.1)	30 (3.9)	1.9 (1.3–2.7)	<0.001
	End stage renal disease	1125 (0.6)	10 (1.3)	2.3 (1.2–4.2)	0.011
	Malignancy				
	Primary (including lymphoma and leukemia)	10,325 (5.3)	66 (8.6)	1.7 (1.3–2.1)	<0.001
	Metastatic	1435 (0.7)	8 (1.0)	1.4 (0.7–2.8)	0.333
	Immunodeficiency	0	0	-	-
	Osteoporosis	35,151 (18.2)	186 (24.3)	1.4 (1.2–1.7)	<0.001

Data are reported as mean ± standard deviation for numerical variables and as numbers and frequencies (%) for categorical variables.

**Table 3 jcm-11-00778-t003:** Comparison of treatment profiles.

Variables	Category	Patients without Postoperative Deep Spine Infection	Patients with Postoperative Deep Spine Infection	Unadjusted Odds Ratio(95% Confidence Interval)	*p*-Value
Surgical approach	Cervical				
	Anterior	42,015 (21.7)	26 (3.4)	0.1 (0.1–0.2)	<0.001
	Posterior	3603 (1.9)	8 (1.0)	0.5 (0.3–1.1)	0.181
	Thoracic				
	Anterior	118 (0.1)	1 (0.1)	2.0 (0.3–14.4)	0.487
	Posterior	4062 (2.1)	28 (3.7)	1.6 (1.1–2.4)	0.012
	Lumbar				
	Anterior	2630 (1.4)	7 (0.9)	0.6 (0.3–1.3)	0.228
	Posterior	96,300 (49.8)	406 (52.9)	reference	
	Multiple approach	44,541 (23.0)	291 (37.9)	1.6 (1.3–1.8)	<0.001
Cage		88,312 (45.7)	424 (55.3)	1.5 (1.3–1.7)	<0.001
Transfusion	Autologous	920 (0.5)	6 (0.8)	1.6 (0.7–3.7)	0.224
	Allogenous	76,784 (39.7)	474 (61.8)	2.5 (2.1–2.8)	<0.001
Systemic steroid	None	118,638 (61.4)	456 (59.5)	reference	<0.001
	Within 2 weeks	67,547 (34.9)	256 (33.4)	1.0 (0.8–1.1)	0.857
	Over 2 weeks	7084 (3.7)	55 (7.2)	2.0 (1.5–2.7)	<0.001
Immunosuppressive agent	None	192,818 (99.8)	766 (99.9)	reference	0.930
	Within 2 weeks	367 (0.2)	1 (0.1)	0.7 (0.1–4.9)	0.707
	Over 2 weeks	84 (0.0)	0	-	0.954

Data are reported as number of cases and frequencies (%) for categorical variables.

**Table 4 jcm-11-00778-t004:** Risk factors for postoperative deep infection after instrumented spinal fusion (Model 1): multivariable analysis with bootstrap validation.

Variables	Category	Model 1			
		Adjusted Odds Rratio (95% Confidence Interval)	*p*-Value	Bootstrap Adjusted Odds Ratio (95% Confidence Interval)	Relative Bias (%)
Age	50–59 vs. 20–49	1.3 (0.9–1.8)	0.235	1.2 (1.0–1.8)	−2.3
	60–69 vs. 20–49	1.6 (1.1–2.2)	0.013	1.6 (1.2–2.1)	6.8
	70–79 vs. 20–49	1.6 (1.1–2.3)	0.008	1.7 (1.3–2.2)	5.4
	80+ vs. 20–49	1.9 (1.3–3.0)	0.003	2.0 (1.4–2.9)	5.6
Sex	Male vs. female	1.8 (1.5–2.1)	<0.001	1.8 (1.6–2.0)	−1.5
Regions	Rural vs. urban	1.3 (1.1–1.5)	0.008	1.3 (1.2–1.5)	14.3
Surgical approach	Cervical anterior vs. lumbar posterior	0.2 (0.1–0.3)	<0.001	0.2 (0.1–0.3)	2.0
	Cervical posterior vs. lumbar posterior	0.4 (0.2–0.9)	0.025	0.4 (0.2–0.7)	7.0
	Thoracic anterior vs. lumbar posterior	1.9 (0.3–13.4)	0.540	0.1 (0.0–5.9)	−557.1
	Thoracic posterior vs. lumbar posterior	1.4 (0.9–2.0)	0.112	1.4 (0.9–1.8)	−2.3
	Lumbar anterior vs. lumbar posterior	0.9 (0.4–1.8)	0.961	0.8 (0.4–1.5)	75.2
	Multiple vs. lumbar posterior	1.3 (1.1–1.5)	<0.001	1.3 (1.2–1.5)	−1.2
Cage		1.1 (0.9–1.3)	0.258	1.1 (0.9–1.3)	5.3
Charlson comorbidity index score	3–5 vs. 0–2	0.9 (0.7–1.2)	0.626	1.0 (0.7–1.4)	−99.8
	over 6 vs. 0–2	1.2 (0.7–2.0)	0.495	1.6 (0.8–2.5)	141.5
Comorbidities	Congestive heart failure	1.1 (0.8–1.5)	0.446	1.3 (0.7–1.5)	141.6
	Cerebrovascular disease	1.5 (1.2–1.8)	<0.001	1.5 (1.3–1.8)	4.9
	Peripheral vascular disease	1.3 (1.0–1.6)	0.017	1.3 (1.2–1.5)	16.3
	Chronic pulmonary disease	1.2 (1.0–1.4)	0.031	1.3 (1.2–1.4)	27.6
	Rheumatologic disease	1.6 (1.2–2.0)	<0.001	1.6 (1.3–1.9)	5.5
	Peptic ulcer	1.0 (0.9–1.2)	0.809	1.1 (0.7–1.3)	202.7
	Liver disease	1.4 (1.1–1.7)	0.003	1.4 (1.2–1.6)	7.7
	Diabetes	1.5 (1.3–1.8)	<0.001	1.5 (1.3–1.8)	−2.2
	Hemiplegia or paraplegia	2.1 (1.4–3.2)	<0.001	2.2 (1.7–3.1)	6.3
	Renal disease	0.9 (0.6–1.5)	0.824	0.9 (0.5–0.7)	74.9
	End stage renal disease	1.5 (0.7–3.0)	0.322	2.0 (1.8–2.6)	90.4
	Malignancy, primary	1.2 (0.8–1.6)	0.352	1.4 (1.3–1.6)	120.6
	Osteoporosis	1.2 (1.0–1.4)	0.075	1.3 (1.2–1.5)	56.9
	Allogenous transfusion	1.5 (1.3–1.8)	<0.001	1.6 (1.4–1.8)	2.5
Systemic steroid	Within 2 weeks vs. no use	1.0 (0.9–1.3)	0.272	1.1 (0.9–1.3)	5.7
	Over 2 weeks vs. no use	1.5 (1.1–2.0)	0.007	1.6 (1.4–2.0)	23.6
Type of hospital	Tertiary vs. others	1.2 (1.0–1.4)	0.111	1.2 (1.0–1.4)	38.6
	General vs. others	1.2 (1.0–1.4)	0.061	1.3 (1.1–1.5)	34.5

All significant independent variables (*p* < 0.05) from the univariate analysis were included in Model 1. Relative bias was estimated as the difference between the mean bootstrapped regression coefficient estimates and the mean parameter estimates of Model 1, divided by the mean parameter estimates of Model 1.

**Table 5 jcm-11-00778-t005:** Risk factors for postoperative deep infection after instrumented spinal fusion (Model 2, final model): multivariable analysis with bootstrap validation.

Variables	Category	Model 2 (backward)			
		Adjusted Odds Ratio (95% Confidence Interval)	*p*-Value	Bootstrap Adjusted Odds Ratio (95% Confidence Interval)	Relative Bias
Age	60–69 vs. 20–49	1.6 (1.1–2.3)	0.007	1.6 (1.2–2.1)	−3.0
	70–79 vs. 20–49	1.7 (1.2–2.5)	0.003	1.7 (1.3–2.2)	−5.5
	80+ vs. 20–49	2.1 (1.3–3.2)	0.001	2.0 (1.4–2.8)	−5.0
Sex	Male vs. female	1.7 (1.5–2.0)	<0.001	1.7 (1.6–2.0)	−0.2
Regions	Rural vs. urban	1.3 (1.1–1.5)	0.005	1.3 (1.1–1.5)	−5.8
Surgical approach	Cervical anterior vs. lumbar posterior	0.2 (0.1–0.3)	<0.001	0.2 (0.1–0.3)	1.6
	Cervical posterior vs. lumbar posterior	0.5 (0.2–1.0)	0.037	0.4 (0.2–0.8)	10.1
	Multiple vs. lumbar posterior	1.4 (1.2–1.6)	<0.001	1.3 (1.2–1.5)	−4.4
Comorbidities	Cerebrovascular disease	1.5 (1.2–1.8)	<0.001	1.5 (1.3–1.8)	−0.7
	Peripheral vascular disease	1.3 (1.1–1.5)	0.009	1.3 (1.1–1.5)	−1.0
	Chronic pulmonary disease	1.2 (1.0–1.4)	0.011	1.2 (1.0–1.4)	−1.0
	Rheumatologic disease	1.6 (1.3–2.1)	<0.001	1.6 (1.3–2.0)	0.3
	Liver disease	1.4 (1.1–1.7)	<0.001	1.4 (1.2–1.6)	−1.8
	Diabetes	1.5 (1.3–1.7)	<0.001	1.5 (1.3–1.7)	1.5
	Hemiplegia or paraplegia	2.2 (1.5–3.3)	<0.001	2.2 (1.5–3.0)	−1.7
Allogenous transfusion		1.6 (1.3–1.8)	<0.001	1.6 (1.4–1.8)	0.2
Systemic steroid	Over 2 weeks vs. no use	1.5 (1.1–2.0)	0.005	1.5 (1.2–1.9)	2.3

All significant independent variables (*p* < 0.05) from the univariate analysis were initially included and subsequently chosen by backward, stepwise selection in Model 2. Relative bias was estimated as the difference between the mean bootstrapped regression coefficient estimates and the mean parameter estimates of Model 2, divided by the mean parameter estimates of Model 2.

## Data Availability

The datasets generated for the current study are not publicly available due to Data Protection Laws and Regulations in Korea, but the analyzing results are available from the corresponding authors on reasonable request.

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
