# Peer review of "Risk Factors for Postoperative Deep Infection after Instrumented Spinal Fusion Surgeries for Degenerative Spinal Disease: A Nationwide Cohort Study of 194,036 Patients"

_jcm, 2022, doi:10.3390/jcm11030778_

Round 1
Reviewer 1 Report
I thank the editor for giving me the opportunity to review your paper. Below, I have provided our comments.
Review of manuscript ID: jcm-1482318
Title
Risk factors for postoperative deep infection after instrumented 2 spinal fusion surgeries for degenerative spinal disease: a nationwide cohort study of 194,036 patients
Outline
This study aimed to fill the gap by assessing the risk factors and their adjusted 13 relative risks through a comprehensive analysis, including all the core information from nationwide, population-based, cohort study.
Critique
#1. Is there a reason for setting the wash-out period to two years? Whether it is a general criterion or whether this period for the above data in other similar studies is generally treated as two years.
#2. Overall, the paper was understandable and the points of spinal infection after instrumentation are well addressed.
Reviewer 2 Report
The authors report a retrospective investigation assessing for risk factors for postoperative deep infection after spinal fusion surgery for degenerative disease using a Korean National Health Insurance claims database. The authors should be commended for their efforts. The large numbers included in the expansive Korean administrative database is a strength. However, the database itself and lack of granularity is the critical limitation.
Many variables are not included that are linked to postoperative infections including:
- Use and type of intra and postoperative antibiotics
- Use of preoperative decontamination protocols.
- Understanding of ongoing quality initiatives that may associate with infection occurrence
- Use and type of Intraoperative topical antibiotics? Use of postoperative drain antibiotic prophylaxis?
- Presence of surgical drain use, number of days drain in place
- Surgical strategy - MIS? Number of levels operated?
- Intraoperative events - durotomy? Number of surgeons? Experience level of surgeons?
- Exact Pathology operated?
- Microbiologic data including positive intraoperative cultures to confirm infection are not available.
Another important limitation is that as a payer-claims-based database, the data is susceptible to an underestimation of SSI. This is supported by the low infection rate of 0.39%. The authors should comment on this further.
For this important lack of granularity, with many factors that could be driving the findings in the authors’ predictor analyses, I do not think the article warrants publication in its current state.
To remedy this in future submissions, the authors should include the above referenced other variables. The authors should proof their article for errors in grammar, which is distracting throughout. Also, in the abstract, several odds ratios are provided but no indicator of confidence interval and/or significance of these ratios are provided. These are necessary for evaluation of their meaning when readers are reviewing the abstract.
Reviewer 3 Report
The authors present an article using a Korean administrative database looking at risk factors for deep wound infection. They found that age, deep vs anterior, steroid use among others are predictive of infection. It is important to understand risk factors for infection - but can the authors provide more about what was novel about this study? Numerous studies have used institutional, multiinstitutional, and administrative databases to predict infection after spine surgery? Where does this study fit in the context of the literature, and what novel conclusions should the readers take from this?
- On a more nuanced point - when the authors provide odds ratio in Tables 3 and 4 comparing treatment approaches - I think it would be more relevant to compare anterior to posterior of the same level. Why would i care about the odds ratio of anterior cervical vs posterior lumbar in a clinical setting? More relevant would be to understand the relative infection risk of treating a pathology in a certain location (ie cervical) comparing the front to the back...
- figure 4 is a helpful summary slide - but when I counsel a patient, how can I use this to guide my treatment approach or best informing patients? Can the authors combine these data into an online calculator or paper nomogram to enable individual-level prediction?
Round 2
Reviewer 2 Report
The authors have adequately addressed the previous concerns.